# Diagnostic Application of Volatile Organic Compounds as Potential Biomarkers for Detecting Digestive Neoplasia: A Systematic Review

**DOI:** 10.3390/diagnostics11122317

**Published:** 2021-12-09

**Authors:** Augustin Catalin Dima, Daniel Vasile Balaban, Alina Dima

**Affiliations:** 1Department of General Surgery and Department of Gastroenterology, Dr. Carol Davila Central Military Emergency University Hospital, 010825 Bucharest, Romania; augustin-catalin.dima@drd.umfcd.ro; 2Carol Davila University of Medicine and Pharmacy, 020021 Bucharest, Romania; 3Department of Rheumatology, Colentina Clinical Hospital, 020125 Bucharest, Romania; alina_dima@outlook.com

**Keywords:** volatile organic compounds, neoplasia, biomarker, breath

## Abstract

Volatile organic compounds (VOCs) are part of the exhaled breath that were proposed as non-invasive breath biomarkers via different human discharge products like saliva, breath, urine, blood, or tissues. Particularly, due to the non-invasive approach, VOCs were considered as potential biomarkers for non-invasive early cancer detection. We herein aimed to review the data over VOCs utility in digestive neoplasia as early diagnosis or monitoring biomarkers. A systematic literature search was done using MEDLINE via PubMed, Cochrane Library, and Thomson Reuters’ Web of Science Core Collection. We identified sixteen articles that were included in the final analysis. Based on the current knowledge, we cannot identify a single VOC as a specific non-invasive biomarker for digestive neoplasia. Several combinations of up to twelve VOCs seem promising for accurately detecting some neoplasia types. A combination of different VOCs breath expression are promising tools for digestive neoplasia screening.

## 1. Introduction

Volatile organic compounds (VOCs) are part of the exhaled breath [1] present in a state of gas at ambient temperature [2]. VOCs might be used as non-invasive biomarkers via different human discharge products like saliva, breath, urine, blood, or tissues [3].

The breath VOCs are components with both exogeneous or endogenous origin responsible for breath characteristics [1]. The VOCs’ cycle includes bloodstream transport after production to the lungs, from where the VOCs are exchanged and exhaled through the breath air [3].

The changes on the breath VOCs composition might confer an unpleasant smell to exhaled breath [1]. The breath particularities have been observed and described since early medicine and are still part of classic clinical semiology, e.g., fetor hepaticus, fetor uremic [1]. However, the first results of gas–liquid partition chromatography were presented only about fifty years ago [4].

Breathomics, the molecular analysis of the exhaled breath, implies the analysis of the human volatilome, a specific unique bio-signature of each individual [2]. The exhaled human breath contains parts of mixed alveolar gas, having nitrogen as the main compound (78%), followed by oxygen (17%), carbon dioxide (4%), argon (< 1%), and water (< 1%) [2]. The VOCs actually represent less than 1% of the exhaled gas [2]. The gas–liquid partition chromatography might quantitatively determine about 250 VOCs in a single sample of breath [4]. The first compendium of all human body VOCs identified in apparently healthy individuals described no less than 1840 different products, of which 872 were breath VOCs [5].

Electronic sensing devices (electronic nose), which have the potential of detecting unique complex mixtures of VOC biomarker metabolites (biomarker mixtures in combination) in human diagnostic samples, and are recorded as unique smell print e-nose signatures, are also under development [6,7,8].

Main metabolic pathways of several VOCs are developed via volatile sulfur compounds (cysteine with production of hydrogen sulfite, tryptophan to indole, lysine to cadaverine, arginine to putrescine, methionine to methyl mercaptan and dimethyl sulfide), via fatty acid with acetone production or via protein metabolism resulting ammonia [1].

Several attempts to define the relationship between specific VOCs and a wide range of systemic diseases have been reported [1]. For some pathologies, VOCs are already used in daily clinical practice. Thus, the 13C-urea measurement is useful for *H. pylori* diagnosis, gastric emptying studies and pancreatic exocrine insufficiency, the hydrogen breath testing is used to determine small intestine bacterial overgrowth, while the exhaled nitric oxide is monitored in asthma [9].

VOCs are considered as potential biomarkers for non-invasive early cancer detection [2,10], as screening tools especially for high risk patients, e.g., smokers, heavy drinkers, or as post-therapeutic monitoring for recurrence occurrence [11]. We herein aimed to review the breath VOCs identified in digestive cancers. To the best of our knowledge, this is the first review focusing on breath VOCs in digestive neoplasia.

## 2. Methods

### 2.1. Search Strategy and Information Sources

The research was conducted according to PRISMA (preferred reporting items for systematic reviews and meta-analysis) guidelines [12]. The literature search was performed without start time-period limitation (up to 1 May 2021) for English language studies, starting with the following medical subject headings (MeSH) terms: “Volatile Organic Compounds” (D055549), “Mouth Neoplasms “ (D009062), “Pancreatic Neoplasms” (D010190), “Liver Neoplasms” (D008113), “Stomach Neoplasms” (D013274), “Esophageal Neoplasms” (D004938), “Duodenal Neoplasms” (D004379), “Colonic Neoplasms” (D003110), “Mouth Neoplasms” (D009062), and “Pharyngeal Neoplasms” (D010610).

The search was performed on MEDLINE via PubMed, Cochrane Library, and Thomson Reuters’ Web of Science Core Collection with the combination of ”Volatile”/“Volatile Organic Compounds” and then, successively, combined with the terms for the digestive neoplasms.

### 2.2. Eligibility Criteria and Study Selection

The studies for this review were selected according to the following criteria: all studies focusing on breath VOCs were included. Articles related to VOCs in other samples (e.g., bile, saliva, gastric tissue) were excluded (see Figure 1). The diagnosis of cancer should have been completed at the patients’ inclusion for breath VOCs analysis. Research reporting breath VOCs utility in anything other than malignant digestive pathology was excluded. Articles referring to electronic devices/electronic nose were also excluded. Two reviewers collected the data independently.

### 2.3. Data Extraction and Quality of the Studies Included 

For each article included in analysis, the following items were extracted: first author’s name, year of publication, the type of digestive cancer, main characteristics of the patients included, the method used for the VOCs assessment, the breath compounds identified and described, the results obtained focusing on the VOCs identified as potential cancer diagnosis or follow-up markers, and the performance of the tests used.

## 3. Results

### 3.1. Publication Selection

For each term assessed—mouth, pharyngeal, esophageal, stomach, duodenal, pancreatic, hepatic, and biliary neoplasm—a number of 29, 18, 42, 63, 3, 19, 137, 10 reports; 1, 0, 0, 4, 0, 2, 3, 1 reports; and 1, 0, 4, 3, 0, 2, 3, 0 reports were identified through database searching on PubMed, Cochrane Library, and Web of Science, respectively (Appendix A). Reports not referring to VOCs or digestive neoplasia, reviews or case reports, and articles not in the English language were excluded after the title and abstract screen. After duplicate removal, a total number of 29 articles were obtained (Appendix A), for which the entire manuscript was screened. 13 articles were excluded out of full texts assessed for eligibility, for the following reasons: different instruments (VOCs searched in samples other than breath), different study design (focusing on electronic devices), or different study populations. Finally, 16 articles were included in qualitative synthesis and the main data were extracted (Table 1; Appendix A).

### 3.2. Breath Sample Collection and Analysis

The analysis of the breath samples was done by gas chromatography coupled with mass spectrometry (GC-MS) [11,13,20,21,22,23], with solid phase microextraction (SPME), [14,18,22,25] or with proton-transfer-reaction time-of-flight mass spectrometry (PTR-ToF-MS), selected ion flow tube mass spectrometry (SIFT-MS) [15], proton reaction mass spectrometry (PTR-MS) [16], or an array of nanosensors [22]. GC-MS retrofitted with electron ionization (EI) [10] or retrofitted with positive chemical ionization (PCI) [10,24]. GC-MS is an off-line analytical technique that needs absorption devices and column calibration for the desired analytes [9], and the VOCs are identified by MS database/library matches [19,21,23]. Instead, SIFT-MS and PTR-MS are able to generate real time VOCs quantification in the exhaled breath [9].

The collection of exhaled breath was done in containers of 500 mL to 3 L [11,14], but oral cavity air, 20 mL, collected in headspace vials, was also analyzed [13]. The VOCs concentration levels in breath samples were compared to that in the mixed breath with headspace concentrations, bronchial breath, gastric-endoluminal air [14], with the highest concentrations being in the gastric-endoluminal air [14].

The breath samples were collected in Mylar bags [11,22], Tedlar bags [13], Nalophan bags [26], steel breath bags [15,23], with an air syringe [24], thermal desorption unit tubes [10], or Tenax tubes [19]. Some authors re-used the bags after cleaning with flowing N2 (99.999% purity) [22].

The first precaution in determining the breath VOCs is to control their origin, endogenous or exogenous. It is especially important to determine the endogenous VOCs and to try to limit any exogenous sources [11,13]. In the research included, different methods used to avoid the patients’ breath sample contamination during collection were described. There was an indication of not eating or drinking at least 2 h [24], 4 h [15,21,23], 6 h [9,11,26], or 12 h (overnight) [19,20,22,25] prior the breath sampling. Because the ambient VOCs enter the human body with breath [16], measurements were done in the laboratory room in order to appreciate the concentrations of the background VOCs [23,24,26].

The sampling for both cases and controls was done in the same room/same clinical environment [19,21,22,26], aerated 30 min before sampling in some reports [21] and with more than one sample collected for each patient [19,22]. The designated VOCs sampling room should not have artificial ventilation [11] and also should not be exposed to cleaning products that might influence the VOCs air composition and alter the sample collected.

Breathed medical air in a suitable mask during the breath sampling was also proposed [21,23]. Some authors included in the sampling protocol the need to avoid any heavy efforts 24 h before collection [19], as well as to rest in the room for 20 min [26] to 1 h before breath sampling. In some studies, the sampling was done before performing invasive digestive evaluation [9,19,26].

In order to obtain accurate sampling, the patients were also asked to gargle three times with 100 mL water [16] or to inhale repeatedly to total lung capacity for 3–5 min through a mouthpiece that contains a filter cartridge on inspiratory port, eliminating so more than 99.99% of exogenous VOCs [22]. In some papers it is also noted that the determinations were done in the morning [24], between 6:00 and 8:00 am [16]. In order to include possible confounders of the patients habits that might modify the volatilome, a short questionnaire on the personal habits [25], including drinking, smoking and drugs used [11,19,24], was applied. Some authors also looked for the co-existence of a *H. pylori* infection [19]. 

Patients with concomitant other site neoplasia or important internal pathologies [13] were not included as controls. All samples should be analyzed within 48 h after collection [23].

Overall, the breath sample collection was described as a non-invasive tool that did not result in significant side effects for the patients included [11].

### 3.3. Synthesis of Results

From the 16 studies included, two researches focused on the VOCs expression in oral carcinoma [11,13], two articles on esophageal cancer [9,16], five articles on esophagogastric cancer [10,14,26], four on gastric cancer [9,18,19,20], two articles on colon neoplasia [21,22], another two on pancreatic adenocarcinoma [23,24], and one focused on hepatocellular carcinoma [25] (Table 1).

About one hundred different VOCs were described in relation to digestive cancers in the articles selected in the present research. Some breath VOCs were identified in association with the neoplasia expression in more than one publication and some VOCs were found to be associated with more than one digestive cancer, as presented in Table 2.

Metabolomics is the study of types of biomarkers which distinguish between different sample types, like healthy controls versus neoplasia, based on differences in concentrations of VOCs present (e.g., higher or lower concentrations compared to controls).

Regarding a possible clinical utility, the breath VOCs analysis showed increased [9,10,14,15,16,17,18,19,20,21,22,23,24,25] or decreased [15,16,21,22,23,24] expression in patients with digestive cancers when compared to controls, increased [11] or decreased [11] levels after surgery for cancer, and also correlations with tumor size [13], tumor degree of differentiation [13], and neoplasia recurrence [13].

VOCs expression in oral squamous cell carcinoma (OSCC) was reported in two papers [11,13]. The first one, by Hartwig et al. [11], found a signature of eight VOCs (methyl ethyl ketone, dibutylhydroxytoluene, N-heptane, toluene, 1-heptene) for OSCC, out of 125 VOCs identified in their research, from which three disappeared after curative surgery, namely dimethyl disulfide, decamethylcyclopenta-siloxane and p-xylene. The second research, Bouza et al. [13], identified 105 VOCs using dedicated libraries for these type of compounds, of which eight VOCs were identified as potential OSCC biomarkers: undecane, dodecane, decanal, benzaldehyde, 3,7-dimethyl undecane, 4,5-dimethyl nonane, 1-octene, hexadecane [13].

Six articles were found referring to VOCs in esophageal [16] or esophagogastric cancers [9,10,14,15,17]. Adam et al. [14] determined the presence of volatile fatty acids (acetic, butyric, pentanoic, hexanoic acids) and acetone in patients with esophagogastric cancers when compared to controls [14], showing higher acetic acid, butyric acid, and pentanoic acid in mixed breath samples from patients compared to controls [14]. Markar et al. [15] conducted a diagnostic validation study, using a five-VOCs diagnostic model: butyric acid, hexanoic acid, decanal, pentanoic acid, and butanal. Chin et al. [10] applied a cross-platform MS annotation in the esophagogastric cancer breathomics for accurate VOCs identification [10]. Zou et al. [16] described seven different ions in the breath mass sample (m/z136, m/z34, m/z63, m/z27, m/z95, m/z107, m/z45) able to distinguish esophageal cancer from healthy subjects [16]. Kumar et al. [26] investigated 17 VOCs in relation to the esophagogastric cancer and identified four VOCs with significant difference between groups: hexanoic acid, phenol, methyl phenol, and ethyl phenol [26]. Additionally, in different research, the authors defined a group of twelve VOCs (pentanoic acid, hexanoic acid, phenol, methyl phenol, ethyl phenol, butanal, pentanal, hexanal, heptanal, octanal, nonanal, decanal) to discriminate from physiological breath samples [9]. The other three articles selected for this review focused only on VOCs as markers for gastric cancer [18,19,20]. Chen et al. [18] showed that the following VOCs can be used to discriminate neoplasia from healthy subjects, but also between early and advanced gastric cancer: isoprene, menthol, pivalic acid, acetone, tetradecane, ethyl-pentane, 3-methylpentane, hexane, hexanol, 2,3-dimethylpentane, 2-methylpentane, 2-methylhexane, dodecane, phenyl acetate [18]. Xu et al. [19] identified as many as 214 VOCs in more than 85% of the samples collected, from which they further described five VOCs (2-propenenitrile, 2-butoxy-ethanol, furfural, 6-methyl-5-hepten-2-one, isoprene) significantly elevated in patients with gastric pathology when compared to healthy controls [19]. Amal et al. [20] analyzed breath samples of almost 100 patients with gastric cancer and identified eight significant VOCS (2-propenenitrile, furfural, 2-butoxy-ethanol, hexadecane, 4-methyloctane, 1,2,3-tri-methylbenzene, α-methyl-styrene, 2-butanone) for gastric cancer [20].

We identified only two publications searching for VOCs as markers in colorectal cancer [21,22]. Thus, De Vietro et al. [21] identified from 29 to 89 VOCs expressed, eleven of them (benzaldehyde, benzene ethyl, benzene methyl, butanoic acid, dodecanoic acid, indole, nonanal, octanoic acid, pentanoic acid, phenol and tetradecane) as the most frequent VOCs in both exhaled breath and ex-vivo neoplastic tissues secretions [21]. Benzaldehyde and indole were increased, while the benzene ethyl expression was decreased in colorectal cancer [21]. Peng et al. [22] searched for VOCs presence in the exhaled breath of patients with lung, breast, prostate, but also colorectal cancer [22]. For colorectal cancer, Peng et al. found increased expression of 1,10-(1-butenylidene)bis benzene, 1-iodo nonane, 2-amino-5-isopropyl-8-methyl-1-azulenecarbonitrile, while decreased levels of 1,3-dimethyl benzene, 1,1-dimethylethyl-thio acetic acid, 4-(4-propylcyclohexyl)-40-cyano[1,10-biphenyl]-4-yl ester benzoic acid were seen in patients with colon cancer when compared to control subjects [22].

Two articles focusing on pancreatic cancer [23,24] and one regarding hepatocellular carcinoma [25] were also found. Markar et al. [23] identified a total of 66 VOCs and further found that formaldehyde, acetone, acetoin, undecane, isopropyl alcohol are increased, while the pentane, N-hexane, 1-butanol, 1-(methylthio)-propane, benzaldehyde, tetradecane, amylene hydrate are decreased in pancreatic cancer [23]. Princivalle et al. [24] found 70 compounds, of which ten were identified to be related to pancreatic cancer, but only five were identified using the available databases. In their paper, M17 ammonia, M43 acetyl group, M71, M74, M89, and M112 were increased, while M34 hydrogen sulfide, M44 acetaldehyde, M62, and M64 sulfur dioxide were decreased in relation to pancreatic adenocarcinoma occurrence [24]. Qin et al. [25] screened three VOCs (3-hydroxy-2-butanone, styrene, decane) as potential biomarkers for hepatocellular carcinoma and showed that 3-hydroxy-2-butanone had the best diagnostic value, without correlation with the alpha-fetoprotein (AFP) levels [25].

### 3.4. Method Performance

Markar et al. [15] showed that, in rather advanced esophagogastric cancers (T3 tumoral stage), for a five-VOCs diagnostic model, and an accuracy with an area under curve (AUC) by receiver operating characteristic (ROC) of 0.850, the sensitivity was 80% and the specificity was 81% [15]. 

Zou et al. [16] described in the breath mass spectrum a combination of seven ions (m/z136, m/z34, m/z63, m/z27, m/z95, m/z107, m/z45) able to distinguish with 86% sensitivity and 90% specificity (AUC 0.943) between esophageal cancer and healthy people [16].

Kumar et al. [26] found an AUC 0.910 for the combination of four VOCs (hexanoic acid, phenol, methyl phenol, and ethyl phenol) [26]. Moreover, using the data of twelve VOCs, significant differences between normal tract versus esophageal and gastric cancer were found, with an AUC of 0.970 and 0.980, respectively [26]. Chen et al. [18] reported 83% sensitivity and 92% specificity to distinguish between healthy and pathological breath samples, after including in the analysis breath levels of fourteen VOCs [18].

Xu et al. [19] reported 89% sensitivity and 90% specificity for gastric cancer versus benign gastric conditions, and 89% sensitivity and 94% specificity for early stage gastric cancer (I and II) versus late stage (III and IV) [19].

Amal et al. [20], based on eight VOCs with significantly different levels between the group with gastric cancer versus the control one, reported the following data over efficacy as diagnostic model: sensitivity 73%, specificity 98%, accuracy 92% [11].

For pancreatic cancer, Markar et al. [23] calculated in the validation cohort an AUC of 0.736 with 81% sensitivity and 58% specificity in differentiating cancer (adenocarcinoma and neuroendocrine tumors) from no cancer and an AUC of 0.744 with 70% sensitivity and 74% specificity for adenocarcinoma versus no cancer [23]. Princivalle et al. [24] measured the concentrations of 92 VOCs in the end-tidal breath and defined a final predictive model based on ten VOCs [24]. For this ten VOCs predictive model, Princivalle et al. [24] obtained very promising results in discriminating pancreatic adenocarcinoma, with an AUC of 0.990, 100% sensitivity and 84% specificity [24]. Qin et al. [25] showed, for the three VOCs (3-hydroxy-2-butanone, styrene, decane) measured by cross-validation, a sensitivity of 83.3% and specificity of 91.7% for discriminating the hepatocellular adenocarcinoma development [25].

## 4. Discussion

Neoplasia remains one of the leading causes of death worldwide and it is estimated that almost half of these deaths could be prevented, especially by early diagnosis [26]. For the digestive tract, the most used screening test is the fecal occult blood test, while invasive endoscopic methods remain the most accurate diagnostic tool, allowing biopsy sampling for histopathologic diagnosis during the same procedure. Regarding biomarkers routinely available in daily practice, there are serum tumor markers like carcinoembryonic antigen (CEA), AFP, carbohydrate antigen (CA) 19-9 or CA15-3 that are not considered specific for diagnosis, but are used for treatment follow-up. 

At present, there is still a need for non-invasive, easy to use, low-cost methods for screening and early detection of digestive cancers. Owing to the promising results of VOCs in various disease diagnoses, including benign gastrointestinal-related pathologies, their analysis was also proposed as an attractive diagnostic tool and a non-invasive marker for different site neoplasia [27,28,29]. 

The volatilome, the volatile fraction of the metabolome, is composed by VOCs produced by the human body that reflects the metabolic processes. Defining the volatilome composition is a domain under development. Regarding the clinical applications, the ^13^C urea breath test for *H. pylori* gastric infection, gastric emptying studies and pancreatic exocrine insufficiency, as well as the hydrogen breath test for bacterial overgrowth and the nitric oxide in asthma, are already used in daily practice with good results [9].

VOCs were recently examined in different neoplasms, like lung, prostate or breast cancer [26]. We herein reviewed the attempt to describe the VOC-specific breath signatures in digestive cancers, an important growing field as the early diagnosis of digestive tract cancers is associated with good and very good survival rates [9].

Schmutzhard et al. [30] published the first study involving VOCs in digestive neoplasia [11], and showed differences between the carcinoma and control group for 42 out of 209 different masses measured in head and neck tumors [30].

The VOCs include a wide range of compounds like hydrocarbons (undecane, nonane, octene), aldehydes (nonanal, decanal, benzaldehyde), esters and alcohols (butyl/octyl acetates, hexanol, benzyl alcohol), and other compounds, such as nitrogen and sulfur-derivate compounds [13]. The hydrocarbons were related to lung inflammation in the lungs [13], while alkanes/alkenes are produced by lipid peroxidation [13]. The alcohol is commonly present by ingestion, either food or drinks, but there is also an endogenous production by hydrocarbon metabolism (cytochrome P450 enzymes) and alcohol dehydrogenase activity [13]. The alcohol is easily determined in the oral cavity, but, on the contrary, it is very difficult to identify it in exhaled breath [13]. Conversely, esters are found in exhaled breath samples and not in oral cavity samples [13]. The nitrogen and sulfur derivate compounds are mainly associated to halitosis and are less reliable as biomarkers for other pathologies [13]. The fatty acids are absorbed in the bowel and might also contribute to carcinogenesis through the cell-membrane production, energy metabolism, cell signaling, and apoptosis prevention [14]. The presence of volatile fatty acids as VOCs is the result of passage from the general circulation across the alveolar-capillary barrier or of the direct release in the digestive or respiratory tract [14].

Different methods used for VOC determination were presented in the articles included above. The GC-MS has long been considered the gold standard method. It allows the separation of individual VOCs based on their physical appearance in mobile and stationary phases from a gas mixture [2]. The SPME is an add-on technique that allows the separation of the compounds separated by the GC column without introducing solvents [2]. There are additional new methods that allow real-time compound identification like PTR-ToF-MS and much better mass resolution due to the deconvolution at peaks. The PTR-MS uses chemical ionization reactions, as the reagent ion is injected into the chamber along with the sample gas for further exothermic proton transfer reaction [2]. The methods using ion mobility are using instruments that are smaller than the classic GC-MS and PTR-ToF-MS, and so are easily transportable [1]. The SIFT-MS uses a chemical ionization reaction under specific thermal conditions for real-time VOC analysis [2]. An important step in analysis is the sampling of breath, which should be done in a CO_2_-controlled manner [31]. Regarding exogeneous compounds, the acetonitrile is the most frequently found in smokers [32], while 2-propanol appears in hospital indoor air [33].

One of the limitations observed for this systematic review is the relatively low number of cases included for each pathology. Most of the data presented are heterogeneous, especially regarding methodology, which even if similar, is not standardized, and the lack of a unified protocol makes extrapolation of results limited. It is of note that the same group of researchers authored more than one of the articles included, which might bias the results. In addition, we noticed difficulties related to the large number of compounds found. Given the limitations of currently available data, delineating frameworks, and conducting further standardized research is needed in order to depict the usefulness and accuracy of VOCs in digestive neoplasia.

## 5. Summary of Evidence

In recent years, the breath has attracted increasing attention from researchers because it is easy to collect and it is a non-invasive tool with great potential for cancer screening, a topic of paramount importance if we consider the increasing numbers of cancers and cancer-related deaths worldwide. For the articles analyzed in this systematic review, there are no individual VOCs proposed as specific cancer biomarkers yet, but several combinations of VOCs seem promising for accurately detecting some neoplasia types. Considering the good specificity and sensibility in discriminating cancer cases from healthy subjects, this kind of alarm VOCs panel might be used soon as screening for early digestive neoplasia. Finally, VOCs proved to be completely non-invasive biomarkers with high acceptability between both patients and physicians.

## Figures and Tables

**Figure 1 diagnostics-11-02317-f001:**
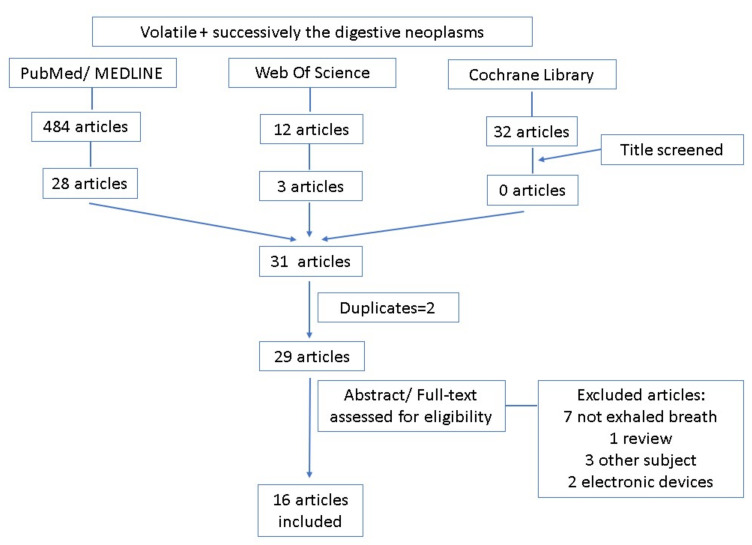
Flow diagram of the systematic literature review process.

**Table 1 diagnostics-11-02317-t001:** Volatile organic compounds in digestive neoplasia: articles included.

No	Author	Pathology	VOCs Proposed as Biomarkers
1	Hartwig et al., 2017 [11]	Oral squamous cell carcinoma (OSCC)	**Signature of eight VOCs for OSCC** **VOCs decreased after cancer surgery:** - Dimethyl disulfide (DDS) - Decamethylcyclopenta-siloxane (D5) - P-xylene (PX) **VOCs increased after cancer surgery:** - Methyl ethyl ketone (MEK) - Dibutylhydroxytoluene (DBH) - N-heptane (NHE) - Toluene (TOL) - 1-heptene (1H)
2	Bouza et al., 2017 [13]	Oral squamous cell carcinoma (OSCC)	**VOCs possible biomarkers for OSCC:** - Undecane - Dodecane - Decanal - Benzaldehyde - 3,7-dimethyl undecane - 4,5-dimethyl nonane - 1-octene - Hexadecane **VOCs correlated with tumor size:** - Benzaldehyde - 3,7-dimethylundecane **VOCs correlated with the histological degree of differentiation:** - Benzaldehyde - Butyl acetate **VOCs correlated with tumor recurrence:** - Benzaldehyde
3	Adam et al., 2019 [14]	Esophagogastric cancer (OGC)	**VFAs increased in mixed breath in esophagogastric cancer:** - Acetic acid - Butyric acid - Pentanoic acid
4	Markar et al., 2018 [15]	Esophagogastric cancer (OGC)	**Five-VOCs diagnostic predictive model:** **increased** - Butyric acid - Hexanoic acid - Decanal **decreased** - Pentanoic acid - Butanal
5	Chin et al., 2018 [10]	Esophagogastric cancer (OGC)	**VOCs predominantly expressed:** - Acetone - Phenol - Benzaldehyde - Butanal
6	Zou et al., 2016 [16]	Esophageal cancer (EC)	**Five ions decreased:** - m/z 34, - m/z 63 Dimethyl sulfide Thioethanol - m/z 95 Phenol 1, 3-cycloheptadiene, - m/z 107 Ethylbenzene P-xylene O-xylene M-xylene benzaldehyde - m/z 45 Acetaldehyde Ethylene oxide **Two ions increased:** - m/z 136, - m/z 27.
7	Kumar et al., 2015 [9]	Esophagogastric cancer (OGC) Esophageal cancer Gastric adenocarcinoma	**VOCs significantly increased in both cancers:** - Pentanoic acid - Hexanoic acid - Phenol - Methyl phenol - Ethyl phenol - Butanal - Pentanal - Hexanal - Heptanal - Octanal - Nonanal - Decanal **Significantly increased in esophageal cancer:** - Butyric acid **No difference between pathologies:** - Methanol - Acetone - Ammonia Isoprene
8	Kumar et al., 2013 [17]	Esophagogastric cancer (OGC)	**Significantly increased in esophageal cancer:** - Hexanoic acid - Phenol - Methyl phenol - Ethyl phenol
9	Chen et al., 2016 [18]	Gastric cancer (GC)	**> 80% of healthy, but barely existed in EGC:** - Isoprene - Menthol **> 80% of GC breath:** - Pivalic acid **> 70% of EGC more than those in the AGC:** - Acetone - Tetradecane **> 65% AGC:** - 2-Methylpentane - 3-methylpentane - Hexane - 2,3-dimethylpentane - 2-methylhexane - 2-methylhexane - Dodecane
10	Xu et al., 2013 [19]	Gastric cancer (GC)	**VOCs increased in gastric cancer:** - 2-propenenitrile - 2-butoxy-ethanol - Furfural - 6-methyl-5-hepten-2-one - Isoprene
11	Amal et al., 2015 [20]	Gastric cancer (GC) Gastric intestinal metaplasia Peptic ulcer disease	**VOCs with significant difference between groups:** - 2-propenenitrile - Furfural - 2- butoxy-ethanol - Hexadecane - 4-methyloctane - 1,2,3-tri-methylbenzene - α-methyl-styrene - 2-butanone
12	De Vietro et al., 2020 [21]	Colorectal cancer	**VOCs increased in colorectal cancer:** - Benzaldehyde - Indole **VOCs decreased in colorectal cancer:** - Benzene ethyl
13	Peng et al., 2010 [22]	Cancers, including colon cancer	**VOCs increased in colon cancer:** - 1,10-(1-butenylidene)bis benzene - 1-iodo nonane - 2-amino-5-isopropyl-8-methyl-1-azulenecarbonitrile **VOCs decreased in colon cancer:** - 1,3-dimethyl benzene - 1,1-dimethylethyl)thio acetic acid - 4-(4-propylcyclohexyl)-40-cyano[1,10-biphenyl]-4-yl ester benzoic acid
14	Markar et al.,2018 [23]	Pancreatic cancer	**VOCs increased in pancreatic cancer:** - Formaldehyde - Acetone - Acetoin - Undecane - Isopropyl alcohol **VOCs decreased in pancreatic cancer:** - Pentane - N-hexane - 1-butanol - 1-(methylthio)-propane - Benzaldehyde - Tetradecane - Amylene hydrate
15	Princivalle et al., 2018 [24]	Pancreatic ductal adenocarcinoma (PDA)	**VOCs increased in PDA:** - M17 ammonia - M43 acetyl group - M71 - M74 - M89 - M112 **VOCs decreased in PDA:** - M34 hydrogen sulfide - M44 acetaldehyde - M62 - M64 sulfur dioxide
16	Qin et al., 2010 [25]	Hepatocellular carcinoma (HCC) Hepatitis Cirrhosis	**VOCs increased in HHC:** - 3-hydroxy-2-butanone - styrene - decane

Abbreviations: GC-MS—gas chromatography mass spectrometry; OLGIM—operative link on gastric intestinal metaplasia; OSCC—oral squamous cell carcinoma; OGC—esophagogastric cancer; MIM—multiple ion monitoring; PUD—peptic ulcer disease; VFAs—volatile fatty acids; VOCs—volatile organic compounds; SIFT-MS—selected ion flow tube mass spectrometry; SPME—gas chromatography mass spectrometry coupled with solid phase microextraction.

**Table 2 diagnostics-11-02317-t002:** Volatile organic compounds associated with digestive neoplasia.

VOCs	Pathology
Acetone	Esophageal cancer [16] Esophagogastric cancer [9,10,14] Gastric cancer [18] Pancreatic cancer [23]
Ammonia	Esophagogastric cancer [9] Pancreatic cancer [24]
Benzaldehyde	Oral carcinoma [13] Esophagogastric cancer [10] Colorectal cancer [21] Pancreatic cancer [23]
Butanal	Esophagogastric cancer [9,10,15]
Butyric acid	Esophageal cancer [9] Esophagogastric cancer [14,15]
Decane/al	Esophagogastric cancer [9,15] Hepatocarcinoma [25]
Dimethyl/hydrogen sulfide	Oral carcinoma [11] Esophageal cancer [10] Pancreatic cancer [24]
Dimethyl/undecane	Oral carcinoma [13] Pancreatic cancer [23]
Ethylbenzene	Esophageal cancer [10] Colon cancer [21]
1,2,3-tri-methylbenzene 1,2-di-methylbenzene	Gastric cancer [20] Colon cancer [22]
Furfural	Gastric cancer [19,20]
Hexanoic acid	Esophagogastric cancer [9,14,15,17]
Hexane/hexanal	Esophagogastric cancer [18] Gastric cancer [18] Pancreatic cancer [23]
Isoprene	Esophagogastric cancer [9] Gastric cancer [18,19]
Phenol	Esophageal cancer [16] Esophagogastric cancer [9,10,17]
Pentanoic Acid	Esophagogastric cancer [9,14]
P-xylene	Oral carcinoma [11] Esophageal cancer [16]
Tetradecane	Gastric cancer [18] Pancreatic cancer [23]

Abbreviations: VOCs—volatile organic compounds.

## Data Availability

Not applicable.

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
