# Peer review of "Diagnostic Application of Volatile Organic Compounds as Potential Biomarkers for Detecting Digestive Neoplasia: A Systematic Review"

_diagnostics, 2021, doi:10.3390/diagnostics11122317_

Round 1

Reviewer 1 Report

The authors have provided a literature review of research studies involved in identifying potential chemical biomarkers of cancers in the GI-tract. The writing requires considerably and extensive revisions to improve English and grammar. The authors should make use of an English-speaking colleague to help in the process of revising the text throughout to make sure sentences do not lack essential words and proper grammar in order to achieve a complete and coherent thought for each statement. There are also other suggestions provided here to improve the manuscript.

Title

Suggested revision of title to better reflect the theme of the paper such as:

Diagnostic Application of Volatile Organic Compounds as Potential Biomarkers for Detecting Digestive Neoplasia: A Systematic Review

Abstract

VOCs are not trace elements. Use of the words "Trace elements" is primarily reserved for inorganic elements (such as zinc and copper) required only in minute amounts by living organisms for normal growth. This mistake was also repeated in the first sentence of "Background".

Remove unnecessary and excess words such as "Further," and "Therefore," and "On the contrary," in all cases throughout [this avoids conversational-type language and states information more concisely according to science, technical-writing convention]

Remove Lines 16-20 (unnecessary because given in detail in the Methods) End the sentence with "Core Collection". [delete words after this for the rest of the sentence] 

1. Background - change word to Introduction

Do not start a new paragraph (first sentence) using a preposition, dependent clause, or adverb (all followed by a comma). This is an indirect statement. First sentences (topic sentences) in each new paragraph should be a direct statement which is much stronger for introducing the topic of the sentence. You can change an indirect statement to a direct statement by moving the preposition, dependent clause or adverb to the end of the sentence and begin the sentence with the subject. This is much more effective writing for topic (first) sentences in new paragraphs.

Line 60 - remove the word "Further"

Consider adding references about electronic-nose devices which have the potential of detecting unique complex mixtures of VOC biomarker metabolites (biomarker mixtures in combination) detected in human diagnostic samples and recorded as unique smellprint e-nose signatures.

Results

Table 1

Column 3 - Patology should be Pathology

Column 4 - marker should be Biomarker

Table 2 not shown

Supplementary Table 3 not displayed for review

Line 113 & 114 - combine references in a sequence e.g. [8, 10-14],

[13, 15-17]

Lines 196-202 the term "metabolomics" needs to be introduced here. These are a type of biomarker which distinguishes between sample types (such as healthy controls vs. disease) based on differences in concentrations of VOCs present (higher or lower, increase or decrease compared to controls etc.).

3.3. Synthesis of Results

Lines 175-195 - make a second Table out of this info and references so it is easier to read

References

Please check to see if there are not more very recent papers (2018-2021) that have come out on this subject within the past few years to make the manuscript very current. Also, consider consulting other recent review articles on disease diagnostics (not necessarily only on the GI tract based on the title) involving analysis of VOCs in human samples. Other reviews may include some GI-tract references, useful to your review, that may not be obvious from just the title of the review. Use search words: noninvasive, VOC biomarker metabolites, volatile disease biomarkers etc.

Author Response

Dear Peer-Reviewer,

First of all, we would like to express our gratitude for the assessment of our manuscript, the valuable suggestions made in your review notes and the opportunity to revise our work. We’ve taken note of your comments and addressed them on a point-by-point basis. Kindly find below our responses and the changes made throughout the manuscript, according to your suggestions.

Comment 1

" Title

Suggested revision of title to better reflect the theme of the paper such as:

Diagnostic Application of Volatile Organic Compounds as Potential Biomarkers for Detecting Digestive Neoplasia: A Systematic Review”.

Author’s reply:

Thank you for this suggestion, we have changed the article’s title according to your proposal.

Comment 2

" Abstract

VOCs are not trace elements. Use of the words "Trace elements" is primarily reserved for inorganic elements (such as zinc and copper) required only in minute amounts by living organisms for normal growth. This mistake was also repeated in the first sentence of "Background".

Remove unnecessary and excess words such as "Further," and "Therefore," and "On the contrary," in all cases throughout [this avoids conversational-type language and states information more concisely according to science, technical-writing convention]

Remove Lines 16-20 (unnecessary because given in detail in the Methods) End the sentence with "Core Collection". [delete words after this for the rest of the sentence]”.

Author’s reply:

Thank you for pointing out the error in the abstract and background sections, also for all the comments about the writing style. We thoroughly checked the manuscript and removed all irrelevant, excessive words.   

Comment 3

"1. Background - change word to Introduction

Do not start a new paragraph (first sentence) using a preposition, dependent clause, or adverb (all followed by a comma). This is an indirect statement. First sentences (topic sentences) in each new paragraph should be a direct statement which is much stronger for introducing the topic of the sentence. You can change an indirect statement to a direct statement by moving the preposition, dependent clause or adverb to the end of the sentence and begin the sentence with the subject. This is much more effective writing for topic (first) sentences in new paragraphs.

Line 60 - remove the word "Further"

Consider adding references about electronic-nose devices which have the potential of detecting unique complex mixtures of VOC biomarker metabolites (biomarker mixtures in combination) detected in human diagnostic samples and recorded as unique smellprint e-nose signatures.”.

Author’s reply:

Thank you again for the observations about the writing style and your help in improving the messages conveyed to the reader throughout the manuscript, we’ve rephrased some of the sentences. We’ve also added a paragraph referring to electronic nose, as suggested.

Comment 4

" Results

Table 1

Column 3 - Patology should be Pathology

Column 4 - marker should be Biomarker

Table 2 not shown

Supplementary Table 3 not displayed for review

Line 113 & 114 - combine references in a sequence e.g. [8, 10-14],

[13, 15-17]

Lines 196-202 the term "metabolomics" needs to be introduced here. These are a type of biomarker which distinguishes between sample types (such as healthy controls vs. disease) based on differences in concentrations of VOCs present (higher or lower, increase or decrease compared to controls etc.).”.

Author’s reply:

Thank you for these observations. Supplementary Files 1, 2, and 3 should be now visible. We’ve made the changes demanded for Table 1 and combined consecutive references. The term “metabolomics” was also introduced, according to your suggestion.

Comment 5

"3.3. Synthesis of Results

Lines 175-195 - make a second Table out of this info and references so it is easier to read"

Author’s reply:

Thank you for this suggestion to gather data in a table. The information presented as text has now been organized as Table 2.

Comment 6

References

Please check to see if there are not more very recent papers (2018-2021) that have come out on this subject within the past few years to make the manuscript very current. Also, consider consulting other recent review articles on disease diagnostics (not necessarily only on the GI tract based on the title) involving analysis of VOCs in human samples. Other reviews may include some GI-tract references, useful to your review, that may not be obvious from just the title of the review. Use search words: noninvasive, VOC biomarker metabolites, volatile disease biomarkers etc.”

Author’s reply:

Thank you for this comment. We’ve added some recent references on disease diagnostics based on VOC analysis, including benign gastrointestinal pathology. If there are specific references which we might have missed and are relevant to the topic, please provide further suggestions.

Thank you again for all your comments, which were highly appreciated and taken into account for this revised, improved version of our manuscript. We hope that all changes made are 

satisfactory resolutions for your inquiries. We remain open to any further corrections.

Best regards,

The authors

Reviewer 2 Report

The presented work contains information gathered from other publications. The authors analyzed and attempted to determine whether there are specific biomarkers from the VOC group for dedicated gastrointestinal diseases. the analysis carried out is correct, but the conclusions drawn are not satisfactory due to the lack of a small amount of data. I think the authors should use one of the data analysis methods (eg PCA) to distinguish which VOC compounds are responsible for a given disease. This would increase the value of the analysis performed.

I think that the topic itself is interesting enough to interest potential readers and it would be worthwhile to proceed with this work.

Author Response

Dear Peer-Reviewer,

First of all, we would like to express our gratitude for the assessment of our manuscript, the valuable suggestions made in your review notes and the opportunity to revise our work. Kindly find below our responses to your comments.

Comment 1

“The presented work contains information gathered from other publications. The authors analyzed and attempted to determine whether there are specific biomarkers from the VOC group for dedicated gastrointestinal diseases. the analysis carried out is correct, but the conclusions drawn are not satisfactory due to the lack of a small amount of data. I think the authors should use one of the data analysis methods (eg PCA) to distinguish which VOC compounds are responsible for a given disease. This would increase the value of the analysis performed.”.

Response

Thank you for this comment, we considered doing supplementary analysis but the information found is insufficient to perform Principal Component Analysis or another method of data analysis. The currently obtained results could be in return important data for future research in order to know what VOCs to consider for digestive neoplasia. Starting from the insufficient data available for further analysis, we’ve added a phrase stating the need for further standardized research on this innovative topic.

Comment 2

“I think that the topic itself is interesting enough to interest potential readers and it would be worthwhile to proceed with this work.”

Response

Thank you for your positive feedback regarding the topic we addresses in our review.

We remain open to any further corrections.

Best regards,

The Authors

Round 2

Reviewer 1 Report

The authors have made a good and thorough effort to make the necessary revisions to correct most of the limitations of the manuscript. There are still a few revisions needed particularly in the Reference section as follows:

All of the abbreviations used in journal titles throughout the Reference section should have periods (as appropriate) for these abbreviations. Please check official abbreviations for each journal and the correct journal formatting for all references cited.

L 411 the journal Bioanalysis should be in italics

L 418 the journal J. Breath Res. - should be in italics

L 465 remove the word (Switzerland)

L 469 the journal name is missing J. Anal. Methods Chem. (in italics); also, remove the words Published Online First: